# The association between Alu hypomethylation and the severity of hypertension

Jirapan Thongsroy[ID]¹*, Apiwat Mutirangura²

**1** School of Medicine, Walailak University, Nakhon Si Thammarat, Thailand, **2** Center of Excellence in Molecular Genetics of Cancer and Human Disease, Department of Anatomy, Faculty of Medicine, Chulalongkorn University, Bangkok, Thailand

* ju_jirapan@hotmail.com

## Abstract

### Introduction

Epigenetic changes that cause genomic instability may be the basis of pathogenic processes of age-associated noncommunicable diseases (NCDs). Essential hypertension is one of the most common NCDs. Alu hypomethylation is an epigenetic event that is commonly found in elderly individuals. Epigenomic alterations are also found in age-associated NCDs such as osteoporosis and diabetes mellitus. Alu methylation prevents DNA from being damaged. Therefore, Alu hypomethylated DNA accumulates DNA damage and, as a result, causes organ function deterioration. Here, we report that Alu hypomethylation is a biomarker for essential hypertension.

### Results

We investigated Alu methylation levels in white blood cells from normal controls, patients with prehypertension, and patients with hypertension. The hypertension group possessed the lowest Alu methylation level when classified by systolic blood pressure and diastolic blood pressure (P = 0.0002 and P = 0.0088, respectively). In the hypertension group, a higher diastolic blood pressure and a lower Alu methylation level were observed (r = -0.6278). Moreover, we found that changes in Alu hypomethylation in the four years of follow-up in the same person were directly correlated with increased diastolic blood pressure.

### Conclusions

Similar to other age-associated NCDs, Alu hypomethylation is found in essential hypertension and is directly correlated with severity, particularly with diastolic blood pressure. Therefore, Alu hypomethylation may be linked with the molecular pathogenesis of high blood pressure and can be used for monitoring the clinical outcome of this disease.

## Introduction

Hypertension is a complex, multifactorial disorder characterized by persistently high blood pressure in the arteries [1–3]. There is increasing empirical evidence that hypertension is one

**Data Availability Statement:** All relevant data are within the paper and its Supporting Information files.

**Funding:** This work was financially supported by Office of the Permanent Secretary, Ministry of

Higher Education, Science, Research and Innovation Grant No. RGNS 63-213 [to J.T.] and 2019 Research Chair Grant from the National Science and Technology Development Agency, Thailand [FDA-CO-2561-8477-TH to A.M.]" The funders had no role in study design, data collection and analysis, decision to publish, or preparation of the manuscript.

**Competing interests:** The authors have declared that no competing interests exist.

of the most serious health problems worldwide [4–6]. A previous study demonstrated that hypertension often has epigenetic alterations, and genome instability is believed to be a contributing factor to hypertension [7–10]. Several studies have reported changes in epigenomic marks and DNA methylation in hypertension and a poor health status in humans [11–16]. Specifically, the level of DNA methylation in hypertensive patients is lower than that in healthy people and depends on the progression of hypertension [17–19]. Moreover, age is a major risk factor for many common noncommunicable diseases (NCDs), such as essential hypertension [20–22].

DNA methylation possesses two main functions: controlling gene regulation and preventing genomic instability [23–27]. Alu methylation preventing DNA damage was reported as one of the mechanisms how DNA methylation prevents genomic instability [28]. The underlying mechanism of this phenomenon is the role of natural occurring hypermethylated DNA gaps in relieving DNA double helix torsional force [29–31]. This mechanism was demonstrated to prevent all kinds of DNA damage [31]. Several reports demonstrated the accumulation of DNA damage in hypertensive patients [32–39]. Therefore, global hypomethylation-driven DNA damage may be part of the pathogenesis of hypertension.

Previously our results showed that Alu element methylation was proven to play a role in preventing the accumulation of endogenous DNA damage [28]. This role may explain the human health deterioration in elderly individuals [28, 40–42]. Meanwhile, our recent study found that Alu methylation in type 2 diabetes mellitus (DM) patients was lower than that in the general population [43]. These results not only showed that there were significantly decreased Alu hypomethylation levels in type 2 DM when compare with normal controls but also that it was associated with high blood pressure in DM patients [43]. We hypothesize that hypomethylation is likely to be observed in patients with hypertension by inducing DNA damage lead to the cellular senescence process and directly correlated with disease severity.

Currently, hypertension is screened by measuring the blood pressure level based on the systolic and diastolic pressure. Several studies have demonstrated that systolic blood pressure is linked to heart attacks and heart failure, as an increase in the diastolic reading can increase the risk of aortic disease, leading to the subsequent degeneration of cellular function in hypertension [44–47]. Recent results found that diastolic pressure was associated with DNA methylation of white blood cell, whereas systolic pressure was not associated with DNA methylation when using Epigenome-wide association study analysis [48]. However, blood pressure is altered with emotional responses to psychosocial stress and following exercise, which does not effectively monitor health deterioration in either geriatric or essential hypertension [49–53]. Therefore, there is a crucial need for additional blood-derived biomarkers of high blood pressure that can be used to monitor the clinical outcome of this disease.

In this study, we used ALU-Combined Bisulfite Restriction Analysis (COBRA) to measure Alu methylation in all of the samples. COBRA-interspersed repetitive sequence PCR is a highly accurate quantitative methylation measurement method, whereas pyrosequencing cannot describe DNA methylation patterns [54–61]. Therefore, in this study, we classified the pattern of Alu methylation in our samples, such as control, prehypertension, and hypertension samples. The purpose of this study was to determine whether Alu methylation may be a highly specific biomarker to predict and prevent the progression of hypertension.

## Materials and methods

### Participants

In the study, we included 240 patients whose blood pressure levels were monitored and were placed them into three groups: normal (123 samples), prehypertension (52 samples) and

hypertension (65 samples). Prehypertension and hypertension patients were admitted to the Tambon Health Promoting Hospital, Thailand, between 2015 and 2019. Patient ages were between 31–85 years. The study was reviewed and approved by the Ethics Clearance Committee on Human Rights Related to Researched Involving Human Subjects, Walailak University, Nakorn Sri Thammarat, Thailand. Written informed consent was obtained from each participant. All subjects voluntarily participated in the study.

## DNA extraction and Bisulfite DNA modification

DNA was extracted from buffy coat by centrifuging whole blood at 1000 x g for 15 minutes at room temperature, and performed DNA extraction with 10% sodium dodecyl sulfate (Sigma Aldrich), lysis buffer II (0.75 M NaCl, 0.024M EDTA at pH 8) and 20 mg/ml proteinase K (USB, OH, USA), and incubated at 50˚C overnight or until cell digestion. Lysed cells were extracted with phenol/chloroform and DNA was precipitated with absolute ethanol [28, 43]. Bisulfite treatment was performed as per standard protocols, with some modifications. Briefly, denatured genomic DNA was incubated in 0.22 M NaOH at 37˚C for 10 min, followed by addition of 30 μl of 10 mM hydroquinone and 520 μl of 3 M sodium-bisulfite for 16–20 h at 50˚C. Subsequently, the DNA was purified and incubated in 0.33 M NaOH at 25˚C for 3 min, ethanol precipitated, washed with 70% ethanol, and resuspended in 20 μl of $H_2O$ [43].

## ALU-Combined Bisulfite Restriction Analysis (COBRA)

These techniques can detect methylated levels of thousands of Cytosine-phosphate-guanine (CpG) loci by using a set of conserved primers for each IRS. To observe the methylation level of Alu in samples, the sodium bisulfite-treated DNA in each sample was amplified by PCR containing 1X PCR buffer (Qiagen, Germany), 0.2 mM deoxynucleotide triphosphate (dNTP) (Promega, USA), 1 mM magnesium chloride (Qiagen, Germany), 25 U of HotStarTaq DNA Polymerase (Qiagen, Germany) and 0.3 μM the primer pair. ALU-Forward 5′-GGYGYGGTG GTTTAYGTTTGTAA-3′, and ALU-Reverse (5′-CTAACTTTTTATATTTTTAATAAAAACRA AATTTCACCA-3′), where R = A and G and Y = C and T. For Alu amplification, the program was set as follows: initial denaturation at 95˚C for 15 minutes followed by 40 cycles of denaturation at 95˚C 45 seconds, annealing at 57˚C 45 seconds, extension at 72˚C 45 seconds, and ending with the final extension at 72˚C 7 minutes. Alu PCR products were subjected to COBRA using 2 U of TaqI (Thermo Scientific, USA), 2 U of TasI (Thermo Scientific, USA) 5X NEB3 buffer (New England Biolabs, USA) and 1 μg/μl bovine serum albumin (BSA) (New England Biolabs, USA).

Each digestion reaction was incubated overnight at 65˚C overnight and later separated on 8% acrylamide and SYBR (Lonza, USA) gel staining. The band intensity of Alu methylation was observed and measured with Phosphoimager using the ImageQuant software (GE Health care, UK) (S1 Fig) [43].

## Methylation analysis

Methylation patterns were classified into four groups based on the COBRA results of the two CpG sites: hypermethylation at both CpGs (mCmC); hypomethylation at CpGs (uCuC); partial methylation of the mCuC; partial methylation of the uCmC (S1 Fig). The Alu methylation levels were calculated for each group based on the intensity of COBRA-digested Alu products. completed. The band intensity of Alu methylation was observed and measured by a Typhoon FLA 7000 and biomolecular imager (GE Health care, UK). In Alu methylation analysis consisted of calculating the band intensity of 5 Alu products sized of 133, 90, 75, 58 and 43 bp

were used for ALU methylation calculations according to the following formula: A = 133/133, B = 58/58, C = 75/75, D = 90/90, E = 43/43, F = [(E + B) − (C + D)]/2. Alu methylation levels were calculated with the following formula: Alu methylation level percentage (%mC) = 100 x (E + B)/(2A + E + B + C + D); percentage of mCmC loci (%mCmC) = 100 x F/(A + C + D + F); percentage of uCmC loci (%uCmC) = 100 x C/(A + C + D + F); percentage of mCuC loci (%mCuC) = 100 x D/(A + C + D + F); and percentage of uCuC loci (%uCuC) = 100 x A/(A+ C + D + F). The same positive control to adjust inter-assay variation [43].

## Statistical analyses

Data were analyzed with SPSS statistical software. The average and distributions of characteristic data of hypertension patients are presented as the mean ± SD and median. *T*-tests were used to determine the differences using a *P* value threshold of 0.05 between the groups in the matched cases based on blood pressure levels. Pearson's correlation coefficient was used to examine the relationship between two continuous variables.

## Results

### Alu methylation in patients with hypertension

First, systolic pressure levels were used to classify the 240 patients into three groups: 75 normal controls, 111 prehypertensive patients, and 54 hypertensive patients. Second, these samples were also classified using diastolic levels, resulting in several patients being placed in different groups (133 normal controls, 69 prehypertensive patients, and 38 hypertensive patients) compared to when the systolic levels were used for classification (Table 1). In this study, we used ALU-Combined Bisulfite Restriction Analysis (COBRA-Alu) to measure Alu methylation in all of the samples. Alu methylation levels of the normal group were compared with those of the prehypertension and hypertension groups; here, we found that the level of Alu methylation was lowest in hypertension (P = 0.0002) (Fig 1A). Similarly, when grouped by the diastolic pressure indicator, we found significantly decreased Alu methylation levels in hypertension patients compared with normal controls (P = 0.0088) (Fig 1B).

**Table 1. Sample size, age, and body mass index (BMI) in each group characterized by the systolic pressure indicator and diastolic pressure indicator.**

| | Group | | | *P value* |
|---|---|---|---|---|
| | **Normal** | **Prehypertension** | **Hypertension** | |
| **Systolic pressure indicator** | | | | |
| *N* | 132 | 54 | 54 | <0.001 |
| Sex | | | | |
| Male | 30 (22.73%) | 11 (20.37%) | 8 (14.81%) | |
| Female | 102 (77.27%) | 43 (79.63%) | 46 (85.19%) | |
| Age (years) (mean±SD) | 53.85 ± 9.47 | 57.39 ± 10.48 | 62.00 ± 11.81 | |
| BMI (kg/m$^2$) (mean±SD) | 25.34 ± 3.64 | 26.81 ± 4.80 | 26.18 ± 5.62 | 0.2431 |
| **Diastolic pressure indicator** | | | | |
| *N* | 173 | 29 | 38 | 0.3902 |
| Sex | | | | |
| Male | 38 (21.97%) | 4 (13.79%) | 7 (18.42%) | |
| Female | 135 (78.03%) | 25 (86.21%) | 31(81.58%) | |
| Age (years) (mean±SD) | 55.92 ± 10.74 | 57.34 ± 11.56 | 58.37± 10.26 | |
| BMI (kg/m$^2$) (mean±SD) | 25.62 ± 4.12 | 26.56 ± 3.12 | 26.40 ± 6.41 | 0.3310 |

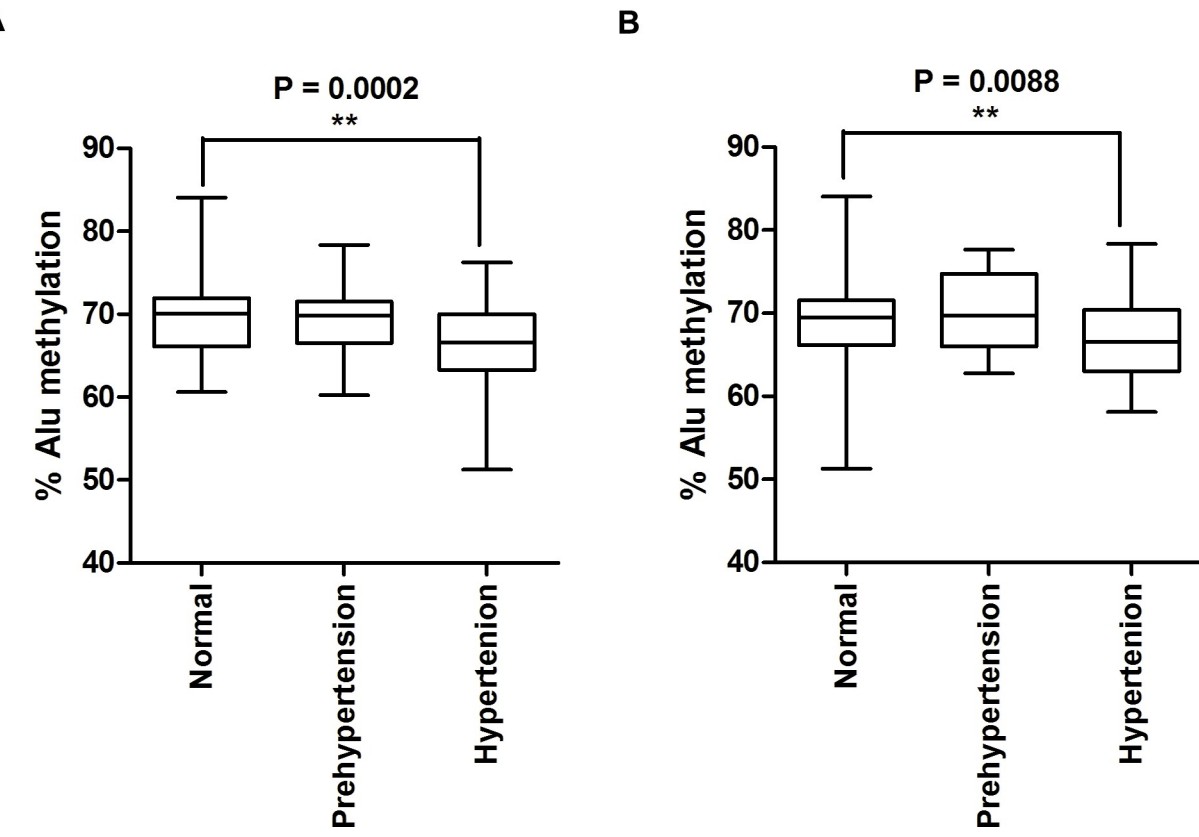

**Fig 1. The percentages of the Alu methylation level in normal controls, prehypertension patients, and hypertension patients.** Three groups were monitored by measuring the systolic and diastolic pressure levels. (A) The Alu methylation pattern is shown for patients grouped based on the systolic pressure level. (B) The Alu methylation show for patients group based on the diastolic pressure level. The percentages of Alu methylation are shown as box plots, with the boxes representing the interquartile ranges (25th to 75th percentile) and the median lines representing the 50th percentile. The whiskers represent the minimum and maximum values. *P<0.05, **P<0.01 (t-test) (Mann–Whitney test).

## Correlation between Alu methylation and blood pressure in the normal, prehypertension and hypertension groups

We examined the correlations between Alu methylation and systolic or diastolic pressure in the normal and hypertension groups (Fig 2). The results revealed significantly inverse correlation between Alu methylation and systolic pressure in the normal and hypertension groups. (Fig 2A and 2C) (r = -0.2144, P = 0.0136 and r = -0.2715, P = 0.0471, respectively). Similarly, a strong inverse association between Alu methylation and diastolic pressure was observed in both normal and hypertension. (Fig 2B and 2D) (r = -0.2186, P = 0.0042 and r = -0.6278, P < 0.0001, respectively).

## Age- and sex-adjusted correlation

To prove whether sex differences affected Alu methylation levels, Alu methylation levels were compared between males and females. Here, we divided males and females into normal, prehypertension, and hypertension groups. We found that Alu methylation levels were not significantly different between males and females when grouping was based on either the systolic indicator (Fig 3A) or diastolic indicator (Fig 3B), similar to previous studies that have shown that sex did not affect Alu methylation [43, 62, 63].

Furthermore, to determine whether the association between Alu methylation and hypertension was influenced by age, we adjusted for age before comparing normal controls and hypertension patients in the groups based on systolic and diastolic pressure (Fig 3C and 3D). To

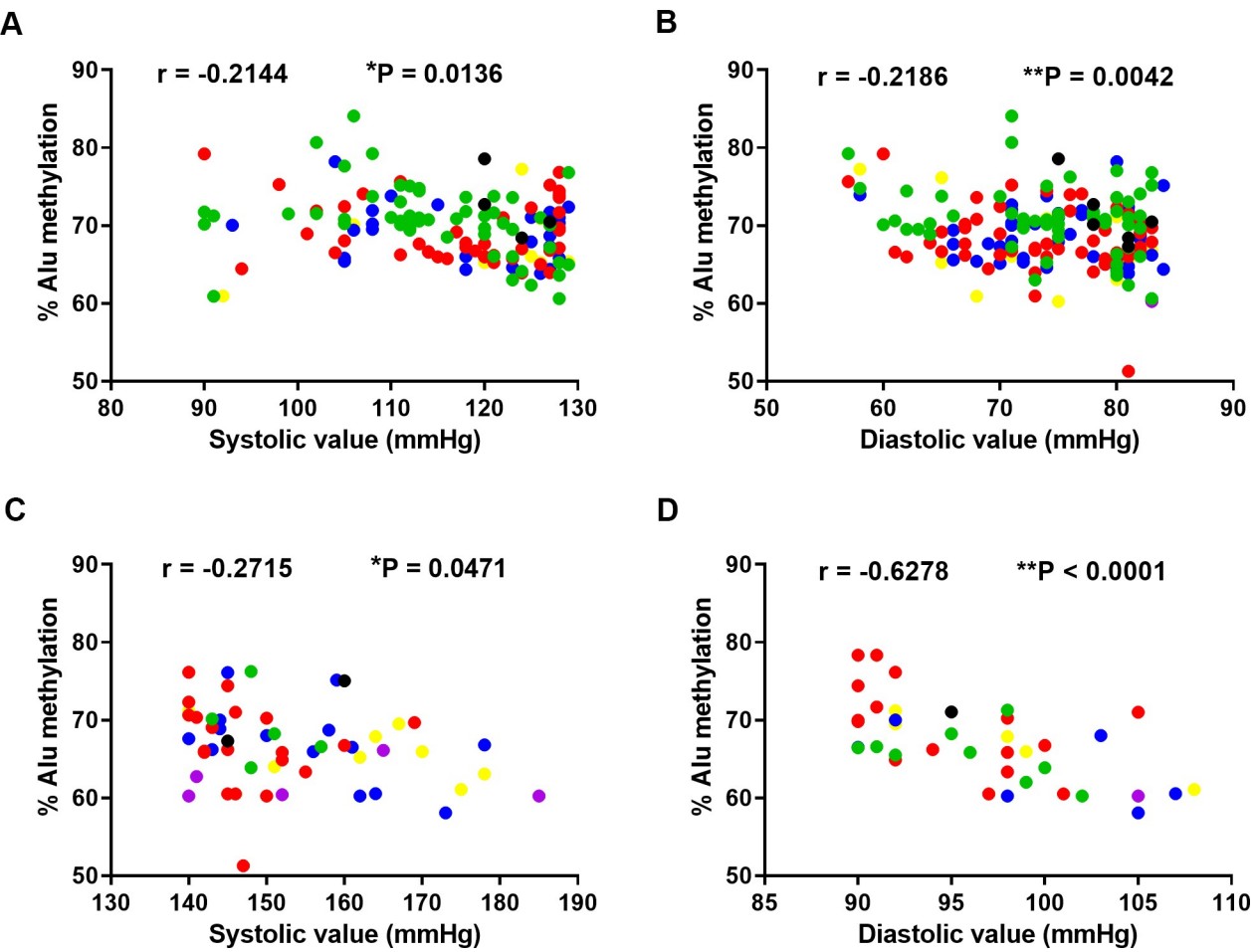

**Fig 2. The association between Alu methylation and blood pressure.** The color representation of the range of age using the black color (31–40 years), green color (41–50 years), red color (51–60 years), blue color (61–70 years), yellow color (71–80 years), and violet color (81–90 years). Correlation between the % Alu methylation and systolic pressure in normal controls (A) and hypertensive patients (C). Correlation between Alu methylation and diastolic pressure in normal controls (B) and hypertension patients (D). Pearson's correlation coefficients (r) with P values are indicated (*P < 0.05, **P < 0.01).

adjust for age, normal samples were matched to same-age hypertension samples to produce 38 age-matched pairs for the systolic indicator and 36 age-matched pairs for the diastolic indicator. The age-matched pairs showed significantly decreased Alu methylation in hypertension patients compared with normal controls when both the systolic (P = 0.0019) and diastolic pressure (P = 0.0122) was used for grouping (Fig 3C and 3D, respectively). Results were similar to findings before age adjustment (Fig 1A and 1B). These results represented that the correlation between Alu hypomethylation and hypertension was not influenced by age.

## Alu methylation levels in hypertensive patients without diabetes and hypertensive patients with diabetes

From normal and hypertension samples, we divided samples into four groups: normal without diabetes, normal with diabetes, hypertensive patients without diabetes, hypertensive patients with diabetes. Using a systolic monitor, we observed that hypertension was significantly lower Alu methylation levels than normal controls in both the non-diabetic and diabetic groups (P = 0.0018 and P = 0446, respectively) (Fig 4A). However, when using the diastolic indicator,

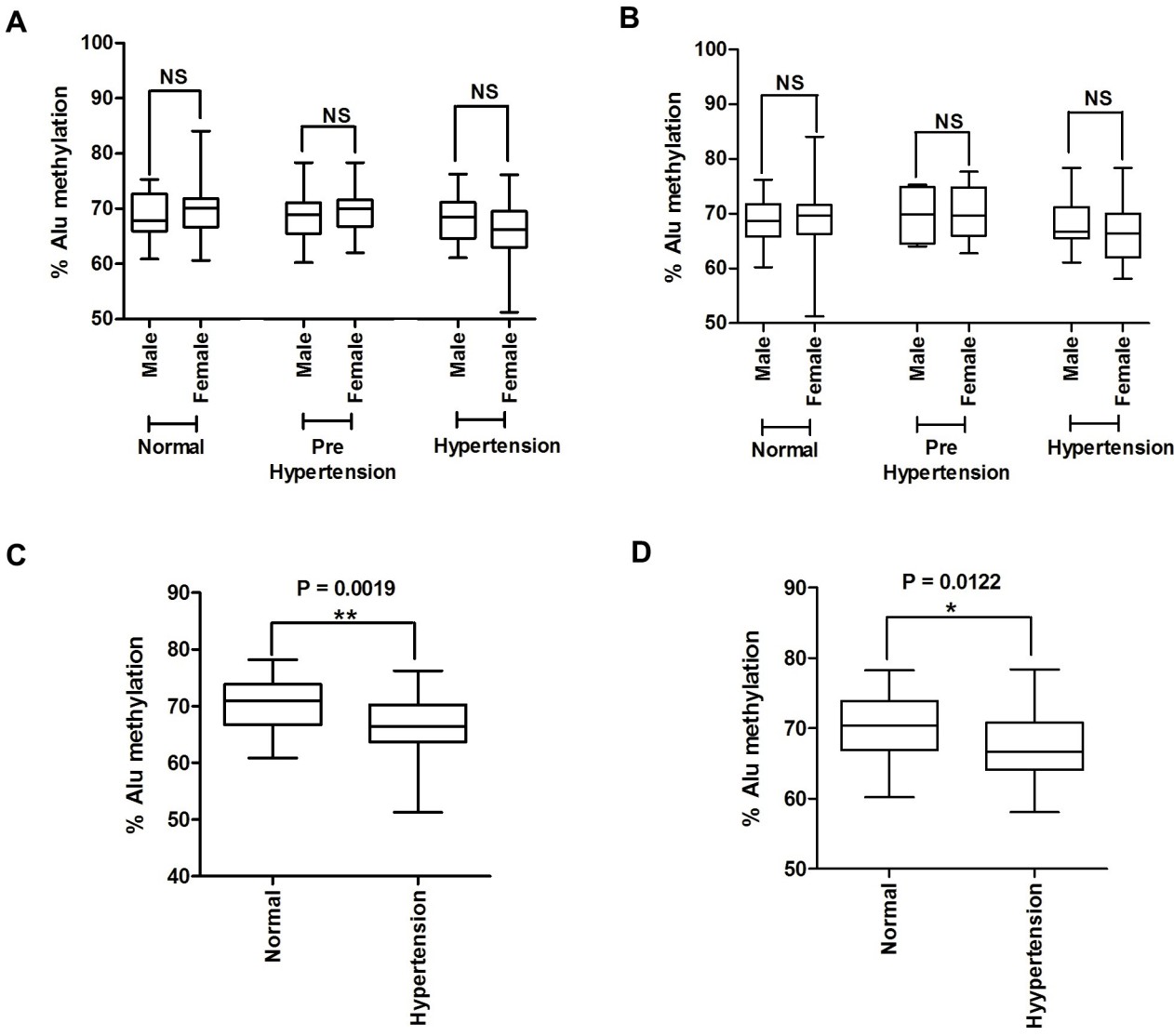

**Fig 3. Alu methylation levels in Age- and sex-adjusted correlation.** Comparisons of Alu methylation levels between males and females in the normal, prehypertension, and hypertension groups when grouping was performed based on the systolic pressure (A) and diastolic pressure indicators (B). The percentage of Alu methylation in age-matched pairs in normal samples and hypertension samples in groups based on both the systolic pressure (C) and diastolic pressure indicators (D). The boxes represent interquartile ranges (25th to 75th percentile), and median lines represent the 50th percentile. The whiskers represent the minimum and maximum values. $^*P < 0.05$, $^{**}P < 0.01$ (Mann–Whitney test).

we found significantly decreased Alu methylation levels in only the non-diabetic group (P = 0.0320) (Fig 4B).

After adjusting the age before compare, the results showed significantly decreased Alu methylation when using the systolic indicator in both the non-diabetic and diabetic groups (P = 0.0016 and P = 0433, respectively) (Fig 4C). When classified by the diastolic pressure, only in non-diabetic group were showed significantly decreased (P = 0.0065) (Fig 4D).

## Alu methylation reduction during case follow-up

Four years (2015–2019) of follow-up in the same person also showed significantly different levels of Alu methylation in the normal and hypertension groups in based on both the systolic indicator (Fig 5A) (P = 0.0122 and P < 0.0001, respectively) and diastolic indicator (Fig 5B)

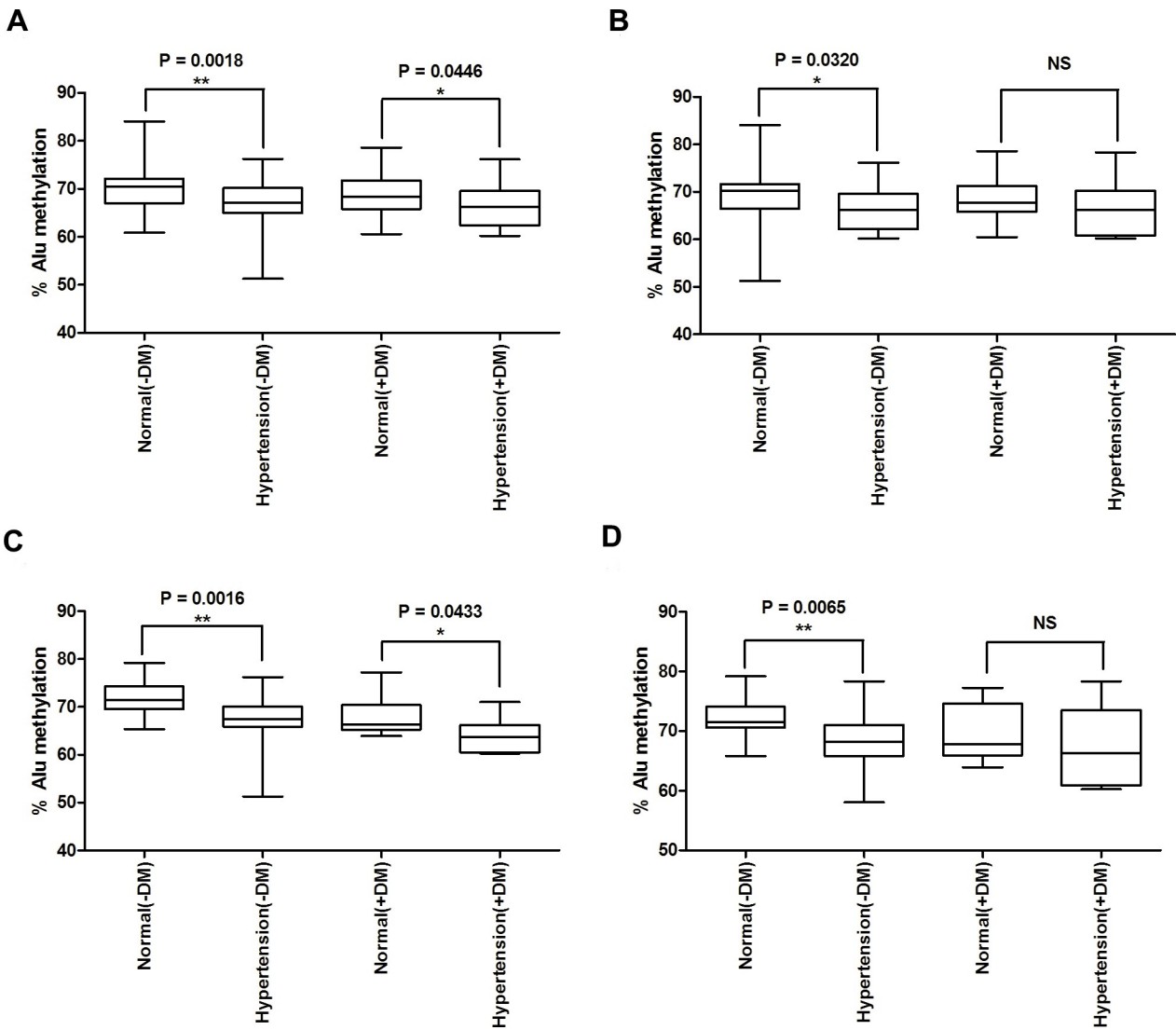

**Fig 4. The percentages of the Alu methylation levels in normal controls and hypertensive patients.** (A) Normal controls without diabetes (Normal(-DM)) compared with hypertension without diabetes (Hypertension(-DM)) and normal controls with diabetes (Normal(+DM)) compared with hypertension with diabetes (Hypertension(+DM)) when classify by the systolic pressure indicator and (B) by the diastolic pressure indicator. (C) In age-matched pairs classify by the systolic pressure indicator and (D) by the diastolic pressure indicator. The percentages of Alu methylation are shown as box plots, with the boxes representing the interquartile ranges (25th to 75th percentile) and the median lines representing the 50th percentile. The whiskers represent the minimum and maximum values. *P<0.05, **P<0.01 (t-test) (Mann–Whitney test).

(P = 0.0040 and P = 0.0066, respectively). An inverse correlation was observed between different levels of Alu methylation in 2019 and 2015 (ΔAlu methylation) and different levels of blood pressure in 2019 and 2015 (Δblood pressure) based on the diastolic indicator (Fig 6B) (r = -0.1608, P = 0.0209). Additionally, a significantly an inverse correlation was observed only between ΔAlu methylation and Δblood pressure in normal individuals when considering the diastolic indicator (Fig 6D) (r = -0.1787, P = 0.0282).

## Discussion

This study found significantly different Alu methylation levels between patients with hypertension and controls when both systolic and diastolic pressure were considered. Importantly, we

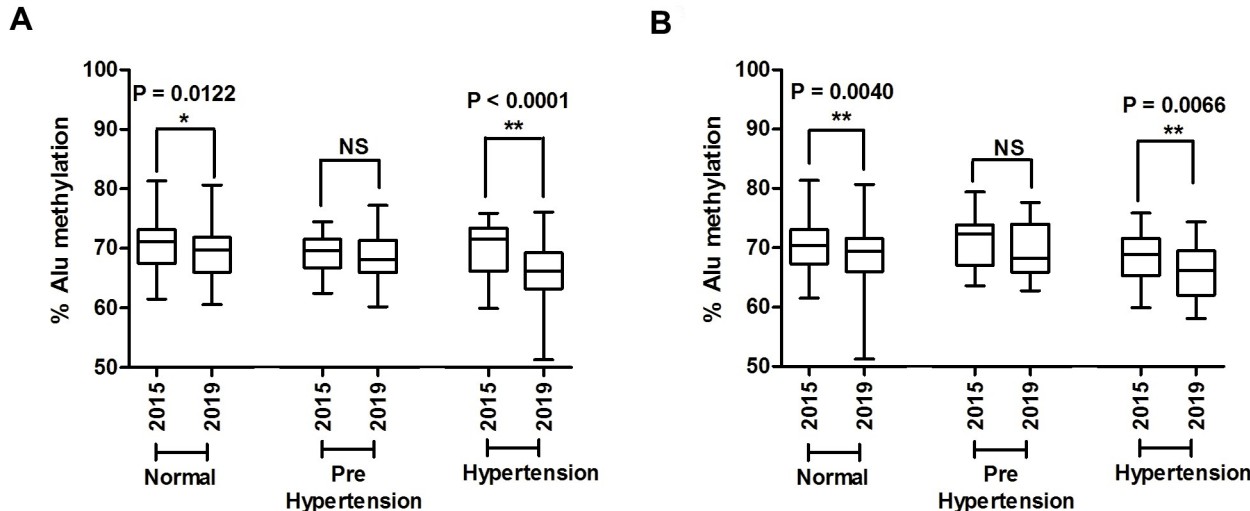

**Fig 5.** Alu methylation levels over 4 years (2015–2019) of follow up in the same person from the normal, prehypertension, and hypertension groups categorized based on the systolic pressure (A) and diastolic pressure indicators (B). *P < 0.05, **P < 0.01 (Paired *t*-test).

found that Alu methylation was an inverse correlation with blood pressure levels in normal controls and hypertensive patients when individuals were classified based on systolic and diastolic blood pressure. This association is similar to the correlation between Alu hypomethylation and lower bone mass or DM in patients [43, 62]. Moreover, Alu hypomethylation is found in hypertensive patients both with and without DM, Alu hypomethylation represents epigenotype which is a key molecular pathogenesis process, of hypertension.

Interestingly, we observed that Alu hypomethylation was significantly different in individuals during follow-up when using systolic and diastolic indicators of hypertension cases and controls. There was no significant difference in prehypertension. Thus, the number of follow-up samples with prehypertension was lower than in the other groups. These limit the use of Alu methylation levels directly as a biomarker for clinical application. The association of Alu hypomethylation with several degenerative diseases involving aging suggests that Alu hypomethylation levels in white blood cells are a promising biomarker for aging phenotypes. Therefore, to determine whether the association between Alu methylation and hypertension was influenced by age, we first adjusted for age before comparing normal and hypertension patients, and the results remained significant after adjusting for age. Second, we observed that ΔAlu hypomethylation levels were significantly an inversely correlated with Δblood pressure when considering diastolic pressure. These results proved the blood pressure changed when Alu hypomethylation changed in the same individuals. Nevertheless, these data did not show an association between ΔAlu hypomethylation levels and Δblood pressure when patients were classified based on systolic pressure. Therefore, our results suggest that decreased Alu methylation was more strongly associated with diastolic pressure than systolic pressure. Similar to a recent experiment that showed an association between diastolic pressure and a lower level of DNA methylation, systolic pressure was not associated with DNA methylation in hypertension [48]. Furthermore, several studies have demonstrated that systolic pressure is more associated with emotional responses to psychosocial stress and exercise than diastolic pressure [49–53]. Moreover, systolic blood pressure is affected by changes in the structure and function of the heart, whereas diastolic pressure can increase the risk of the progression of arterial diseases that cause arterial aging in hypertension [45–47, 64–68].

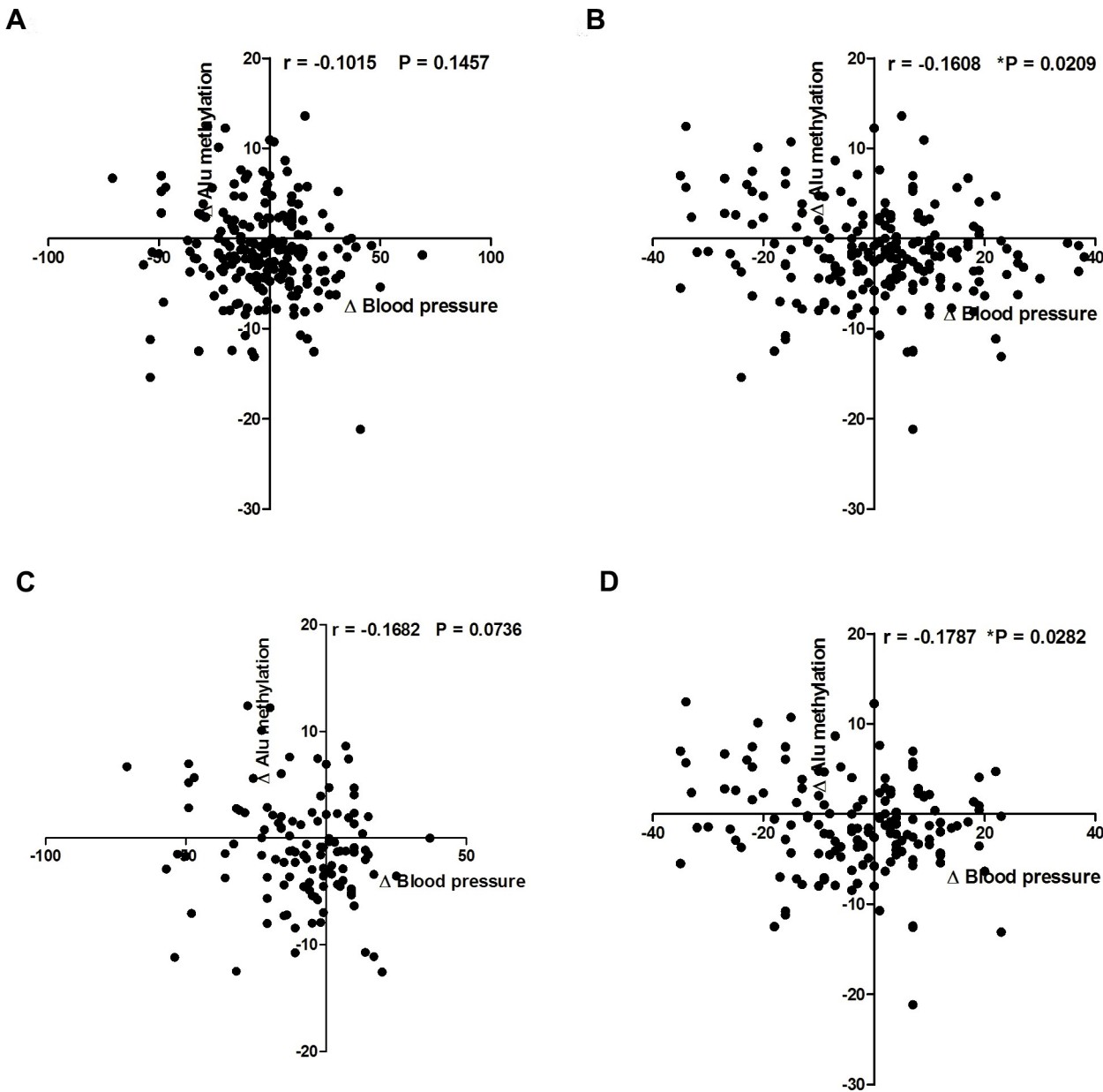

**Fig 6.** Correlation between the levels of ΔAlu methylation and Δblood pressure when using the systolic indicator (A) all of sample (normal, prehypertension, and hypertension), (C) only normal sample. And when using diastolic indicator (B) all of sample (normal, prehypertension, and hypertension), (D) only normal sample. Each plot shows ΔAlu methylation levels in the sample. Pearson's correlation coefficients (r) with P values are indicated (*P < 0.05, **P < 0.01).

The association between Alu hypomethylation and diastolic blood pressure may indicate that the cellular senescence of the artery is involved in the pathogenesis of high blood pressure. Alu hypomethylation increases the accumulation of endogenous DNA damage [28]. The persistence of the DNA damage response transforms cells to the senescent state [69–71]. Here, we propose two hypotheses. First, Alu hypomethylation may be found in all tissues, including arteries. Senescent cells have limited physiological function and promote inflammation, and consequently cause atherosclerosis. Second, Alu hypomethylation in circulating mononuclear

cells may increase DNA damage and consequently limit their capability in the arterial tissue regeneration process.

Hypertension is currently screened by measuring the blood pressure level, and there is no effective biomarker to monitor health that deteriorates in either geriatric or essential hypertension [52, 53]. Similar to hemoglobin A1c (HbA1C) in DM, which is an effective biomarker [72], Alu methylation levels could provide an overall picture of the average disease severity over a long period when detecting hypertension. Thus, cellular senescence was proposed to be one of the cellular dysfunction that promoted pathogenic mechanisms of hypertension [73], as one known mechanism of cellular aging was reduced methylation levels promoting genomic instability [74, 75]. Our previous publication demonstrated that Alu hypomethylation was associated with increasing endogenous DNA damage. Additionally, Alu small interfering RNA (siRNA) could increase Alu methylation, decrease endogenous DNA damage, and increase cell viability [28]. In this study, we observed Alu hypomethylation in hypertensive patients. So, Alu hypomethylation accumulating DNA damage may lead to cellular aging vessels of hypertensive patients. Here, we hypothesize that Alu siRNA might increase Alu methylation, resulting in decreased genomic instability and reduced blood pressure status in hypertensive patients. Therefore, Alu methylation-editing technology, such as Alu siRNA, may play a role in the treatment of hypertensive patients.

## Conclusions

Our results showed significant Alu hypomethylation levels in hypertension patients compared with normal controls and observed an inverse correlation between Alu methylation and blood pressure levels in hypertensive patients. Alu hypomethylation may promote DNA damage, leading to the deterioration of cellular function in hypertensive patients. Therefore, Alu methylation is a promising biomarker for monitoring hypertension and could be considered a target for the future development of better therapeutic methods for prevention and treatment.

## Supporting information

**S1 Fig. COBRA-Alu assay and methylation patterns of Alu amplicons.**
(PDF)

**S2 Fig. The original blot and gel image.**
(PDF)

## Acknowledgments

We thank the staff of the Ban Ton Riang Tambon Health Promoting Hospital, Nakorn Sri Tummarat, Thailand for their assistance in sample collection.

## Author Contributions

**Conceptualization:** Jirapan Thongsroy, Apiwat Mutirangura.

**Data curation:** Jirapan Thongsroy.

**Formal analysis:** Jirapan Thongsroy.

**Funding acquisition:** Jirapan Thongsroy, Apiwat Mutirangura.

**Investigation:** Jirapan Thongsroy.

**Methodology:** Jirapan Thongsroy.

**Project administration:** Jirapan Thongsroy.

**Resources:** Jirapan Thongsroy.

**Software:** Jirapan Thongsroy.

**Supervision:** Jirapan Thongsroy, Apiwat Mutirangura.

**Validation:** Jirapan Thongsroy, Apiwat Mutirangura.

**Visualization:** Jirapan Thongsroy, Apiwat Mutirangura.

**Writing – original draft:** Jirapan Thongsroy, Apiwat Mutirangura.

**Writing – review & editing:** Jirapan Thongsroy, Apiwat Mutirangura.

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
