## [Decision Letter · Decision Letter 0]

26 Apr 2022

PONE-D-21-37190

The association between Alu hypomethylation and the severity of hypertension

PLOS ONE

Dear Dr. Thongsroy,

Thank you for submitting your manuscript to PLOS ONE. After careful consideration, we feel that it has merit but does not fully meet PLOS ONE’s publication criteria as it currently stands. Therefore, we invite you to submit a revised version of the manuscript that addresses the points raised during the review process.

We look forward to receiving your revised manuscript.

Kind regards,

H. Hakan Aydin, MD, FAACC

Academic Editor

PLOS ONE

“This s was financially supported by Office of the Permanent Secretary, Ministry of Higher Education, Science, Research and Innovation Grant No. RGNS 63-213 [to J.T.] and 2019 Research Chair Grant from the National Science and Technology Development Agency, Thailand [FDA-CO-2561-8477-TH to A.M.]”

4. Thank you for stating the following in the Funding Section of your manuscript:

“This s was financially supported by Office of the Permanent Secretary, Ministry of Higher Education, Science, Research and Innovation Grant No. RGNS 63-213 and 2019 Research Chair Grant from the National Science and Technology Development Agency, Thailand [FDA-CO-2561-8477-TH to A.M.]”

“This s was financially supported by Office of the Permanent Secretary, Ministry of Higher Education, Science, Research and Innovation Grant No. RGNS 63-213 [to J.T.] and 2019 Research Chair Grant from the National Science and Technology Development Agency, Thailand [FDA-CO-2561-8477-TH to A.M.]”

6. PLOS requires an ORCID iD for the corresponding author in Editorial Manager on papers submitted after December 6th, 2016. Please ensure that you have an ORCID iD and that it is validated in Editorial Manager. To do this, go to ‘Update my Information’ (in the upper left-hand corner of the main menu), and click on the Fetch/Validate link next to the ORCID field. This will take you to the ORCID site and allow you to create a new iD or authenticate a pre-existing iD in Editorial Manager. Please see the following video for instructions on linking an ORCID iD to your Editorial Manager account: https://www.youtube.com/watch?v=_xcclfuvtxQ.

Additional Editor Comments:

There are essential points that the reviewers' raised in their reviews, including presenting additional data and explaining the methodology and conclusion, and making significant revisions to the text.

Reviewers' comments:

Reviewer's Responses to Questions

**Comments to the Author**

1. Is the manuscript technically sound, and do the data support the conclusions?

Reviewer #1: Yes

Reviewer #2: Partly

2. Has the statistical analysis been performed appropriately and rigorously? 

Reviewer #1: Yes

Reviewer #2: I Don't Know

3. Have the authors made all data underlying the findings in their manuscript fully available?

Reviewer #1: Yes

Reviewer #2: Yes

4. Is the manuscript presented in an intelligible fashion and written in standard English?

Reviewer #1: Yes

Reviewer #2: Yes

5. Review Comments to the Author

Reviewer #1: Overall a nice study on a clinically relevant topic. The manuscript requires further clarification on certain areas (see below). Overall it is a well written paper.

Ln41-42; differentiate between genetically inherited risk and epigenetic alterations over time (is there a trans-generational risk conferred of epigenetic alterations?). Sentence is ambiguous.

Denote and define abbreviations before use in text of manuscript.

How was blood pressure measured- was it 24hour ambulatory blood pressure, or static (if so when was it measured and was this kept constant).

Materials and Methods;

This section requires editing and improvement. The reader must be well informed of the methods used and how they were undertaken

e.g. How were the white blood cells isolated or were they (whole blood used?). Were leucocytes immuno-isolated by magnetic separation? Better description of the BP analysis needed and classification of cohorts. Were CBCs and blood indices recorded? Better description of Alu methylation calculations required also reference needed for the original paper(s) describing the technique.

Ln114-116; syntax requires revising

Ln152-153; error

Ln178; error

Ln179; error

Reviewer #2: The manuscript by Thongsroy and Mutirangura investigates the interesting possibility of an association between hypertension and hypomethylation of Alu repeats. The authors are not sufficiently careful in their scientific statements and their methodology is unconvincing.

Abstract. Alu methylation cannot prevent DNA damage unlike what is stated. There are very many types of DNA methylation which have no relationship to Alu repeats. In addition, the authors need to distinguish in the Abstract between unrepaired or persisting DNA damage and DNA not being accessible to certain types of DNA damage. It is essential that age be mentioned in the Abstract.

Lines 50 -52. Association does not indicate causality. “Therefore” should be deleted.

Line 56. “Significant” should be used only for a statistical test or modified by “biologically.” Do they mean “There was signifant hypomethylation in DM patients compared to ….”

Line 66. Whenever the authors refer to epigenetics or transcription, the type of tissue or cell must be specied and for DNA methylation the type of analysis (overall levels, methylation at specific gene promoters, etc.).

Line 137. Figures 1, 2, 4. All data should be age adjusted, not just some of it. All comparisons (not just some) should be between individuals of the same age group unless the effect of age is being determined. It is well established that DNA methylation levels change with age. The authors should have separately analyzed the age effect among their normal controls.

Line 246. The authors should be much more careful in their scientific statements. What is “improve Alu methylation”? How could medicine target Alu methylation in blood vessels? There are so many sources of changes in blood pressure that do not involve DNA damage that the authors are ignoring.

Methods. What evidence do they have that TaqI digestion was always complete? This is important because of the small differences they are interpreting.

They need technical duplicates or triplicates to show no differences.

They should display a stained gel so that the reader can see the resolution obtained for these small bands.

6. PLOS authors have the option to publish the peer review history of their article (what does this mean?). If published, this will include your full peer review and any attached files.

Reviewer #1: **Yes: **Ronan Murphy

Reviewer #2: No

---

## [Author Response · Author response to Decision Letter 0]

4 May 2022

Dear Editor,

 We appreciate the reviewers’ comments and have revised our manuscript accordingly. We list our responses in a point-by-point fashion and all the changes in the manuscript text are highlighted in yellow.

Reviewer #1: Overall a nice study on a clinically relevant topic. The manuscript requires further clarification on certain areas (see below). Overall it is a well written paper.

Reply: Thank you very much.

Ln41-42; differentiate between genetically inherited risk and epigenetic alterations over time (is there a trans-generational risk conferred of epigenetic alterations?). Sentence is ambiguous.

Reply: Thank you very much. Here in this version, we have edited the word form “inherited genetic component” to “epigenetic alterations” (page 2 line 42)

Denote and define abbreviations before use in text of manuscript.

Reply: We would like to thank reviewer 1 for suggesting additionally defined abbreviations in our manuscript. In this version, we added more defined abbreviations in this manuscript: diabetes mellitus (DM), Combined Bisulfite Restriction Analysis (COBRA), Cytosine-phosphate-guanine (CpG), hemoglobin A1c (HbA1C), and small interfering RNA (siRNA). (page 2 line 60, page 3 line 77, page 4 line 109-110, page 9 line 259 and 265, respectively)

How was blood pressure measured- was it 24hour ambulatory blood pressure, or static (if so when was it measured and was this kept constant).

Reply: Thank you very much. The participants in this study were chosen by the family care team, and blood pressure was monitored weekly to keep constant levels. Furthermore, we also took the blood pressure of the sample for an hour before beginning our experiment.

Materials and Methods

This section requires editing and improvement. The reader must be well informed of the methods used and how they were undertaken

e.g. How were the white blood cells isolated or were they (whole blood used?). Were leucocytes immuno-isolated by magnetic separation? Better description of the BP analysis needed and classification of cohorts. Were CBCs and blood indices recorded? Better description of Alu methylation calculations required also reference needed for the original paper(s) describing the technique.

Reply: Thank you very much for this valuable advice. We updated the Materials and Methods to explain that DNA was extracted from the buffy coat by centrifuging whole blood, using lysis buffer and proteinase K digestion, and phenol-chloroform extraction protocols. Please see the DNA extraction and Bisulfite DNA modification topic in the Materials and Methods (page 4 line 97-101). Furthermore, the reference for the description of Alu methylation calculations has already been added. (page 5 line 142) 

Ln114-116; syntax requires revising

Reply: We apologize for this mistake. The sentence has been edited from "COBRA results were categorized into four groups based on the status of the methylation at the two CpG dinucleotides” to “Methylation patterns were classified into four groups based on the COBRA results of the two CpG sites. (page 5 line 129-131) 

Ln152-153; error

Reply: We apologize for this mistake. The word has been edited from “a significantly an inverse” to “significantly inverse” (page 6 line 168) 

Ln178; error

Reply: We apologize for this mistake. The word has been edited from “hypertensiove” to “hypertensive” (page 7 line 196) 

Ln179; error

Reply: We apologize for this mistake. The word has been edited from “were” to “was” (page 7 line 197) 

Reviewer #2: The manuscript by Thongsroy and Mutirangura investigates the interesting possibility of an association between hypertension and hypomethylation of Alu repeats. The authors are not sufficiently careful in their scientific statements and their methodology is unconvincing.

Abstract. Alu methylation cannot prevent DNA damage unlike what is stated. There are very many types of DNA methylation which have no relationship to Alu repeats. In addition, the authors need to distinguish in the Abstract between unrepaired or persisting DNA damage and DNA not being accessible to certain types of DNA damage. It is essential that age be mentioned in the Abstract.

Reply: Thank you very much, Alu methylation preventing DNA damage was reported as one of the mechanisms how DNA methylation prevents genomic instability (Patchsung, Settayanon et al. 2018). The underlying mechanism of this phenomenon is the role of natural occurring hypermethylated DNA gaps in relieving DNA double helix torsional force (Yasom, Watcharanurak et al. , Pornthanakasem, Kongruttanachok et al. 2008, Thongsroy, Patchsung et al. 2018). This mechanism was demonstrated to prevent all kinds of DNA damage (Yasom, Watcharanurak et al.). Therefore, In this version, we added this paragraph in the introduction (page 2 line 50-54)

References

1. Patchsung, M., et al., Alu siRNA to increase Alu element methylation and prevent DNA 

 damage. Epigenomics, 2018. 10(2): p. 175-185.

2. Pornthanakasem, W., et al., LINE-1 methylation status of endogenous DNA double-strand 

 breaks. Nucleic acids research, 2008. 36(11): p. 3667-3675.

3. Thongsroy, J., et al., Reduction in replication‐independent endogenous DNA double‐strand 

 breaks promotes genomic instability during chronological aging in yeast. The FASEB 

 Journal, 2018. 32(11): p. 6252-6260.

4. Yasom, S., et al., The roles of HMGB1‐produced DNA gaps in DNA protection and aging 

 biomarker reversal. FASEB BioAdvances.

Lines 50 -52. Association does not indicate causality. “Therefore” should be deleted.

Reply: Thank you very much. Here in this version, we added the mechanisms supporting the Alu methylation preventing DNA damage in the introduction. (page 2 line 50-54)

Line 56. “Significant” should be used only for a statistical test or modified by “biologically.” Do they mean “There was signifant hypomethylation in DM patients compared to ….”

Reply: Thank you very much. The sentence was edited from “there were significant Alu hypomethylation levels in type 2 DM” to “there were significantly decreased Alu hypomethylation levels in type 2 DM when compare with normal controls” (page 3 line 62)

Line 66. Whenever the authors refer to epigenetics or transcription, the type of tissue or cell must be specied and for DNA methylation the type of analysis (overall levels, methylation at specific gene promoters, etc.).

Reply: Thank you very much. Here in this version, we have added the type of cell and type of analysis in the sentence “Recent results found that diastolic pressure was associated with DNA methylation of white blood cell, whereas systolic pressure was not associated with DNA methylation when using Epigenome-wide association study analysis. (page 3 line 71-72)

Line 137. Figures 1, 2, 4. All data should be age adjusted, not just some of it. All comparisons (not just some) should be between individuals of the same age group unless the effect of age is being determined. It is well established that DNA methylation levels change with age. The authors should have separately analyzed the age effect among their normal controls.

Reply: Thank you very much. Fig 3 was the graph after the adjusted age of Fig 1, and Figs 4C and 4D were the graphs after the adjusted age of Figs 4A and 4B, respectively. Figure 2 shows the correlation graph. In this version, we included age information in Figure 2 by adding the colors to distinguish different age groups

Line 246. The authors should be much more careful in their scientific statements. What is “improve Alu methylation”? How could medicine target Alu methylation in blood vessels? There are so many sources of changes in blood pressure that do not involve DNA damage that the authors are ignoring.

Reply: Thank you very much. In this version, we added previous evidence supporting Alu hypomethylation might promote accumulated DNA damage leading to genomic instability by loss of cellular aging function in patients with hypertension in the discussion. (page 9 line 264-272)

“Our previous publication demonstrated that Alu hypomethylation was associated with increasing endogenous DNA damage. Additionally, Alu small interfering RNA (siRNA) could increase Alu methylation, decrease endogenous DNA damage, and increase cell viability (Patchsung, Settayanon et al. 2018). In this study, we observed Alu hypomethylation in hypertensive patients. So, Alu hypomethylation accumulating DNA damage may lead to cellular aging vessels of hypertensive patients. Here, we hypothesize that Alu siRNA might increase Alu methylation, resulting in decreased genomic instability and reduced blood pressure status in hypertensive patients. Therefore, Alu methylation-editing technology, such as Alu siRNA, may play a role in the treatment of hypertensive patients.”

Reference

1. Patchsung, M., et al., Alu siRNA to increase Alu element methylation and prevent DNA 

 damage. Epigenomics, 2018. 10(2): p. 175-185.

evidence do they have that TaqI digestion was always complete? This is important because of the small differences they are interpreting.

They need technical duplicates or triplicates to show no differences.

They should display a stained gel so that the reader can see the resolution obtained for these small bands.

Reply: Thank you very much. The below figure shows the duplicate samples that showed no differences. The same patterns of the samples were digested by TaqI digestion. Furthermore, we used the same positive control to adjust inter-assay variation. (page 5 line 142). In this version, we have already added this stained gel figure in Supplement Figure 1B (S1 fig).

---

## [Decision Letter · Decision Letter 1]

23 May 2022

PONE-D-21-37190R1The association between Alu hypomethylation and the severity of hypertensionPLOS ONE

Dear Dr. Thongsroy,

Thank you for submitting your manuscript to PLOS ONE. After careful consideration, we feel that it has merit but does not fully meet PLOS ONE’s publication criteria as it currently stands. Therefore, we invite you to submit a revised version of the manuscript that addresses the points raised during the review process.

We look forward to receiving your revised manuscript.

Kind regards,

H. Hakan Aydin, MD, FAACC

Academic Editor

PLOS ONE

Reviewers' comments:

Reviewer's Responses to Questions

**Comments to the Author**

1. If the authors have adequately addressed your comments raised in a previous round of review and you feel that this manuscript is now acceptable for publication, you may indicate that here to bypass the “Comments to the Author” section, enter your conflict of interest statement in the “Confidential to Editor” section, and submit your "Accept" recommendation.

Reviewer #1: All comments have been addressed

Reviewer #3: (No Response)

2. Is the manuscript technically sound, and do the data support the conclusions?

Reviewer #1: Yes

Reviewer #3: Yes

3. Has the statistical analysis been performed appropriately and rigorously? 

Reviewer #1: Yes

Reviewer #3: Yes

4. Have the authors made all data underlying the findings in their manuscript fully available?

Reviewer #1: Yes

Reviewer #3: Yes

5. Is the manuscript presented in an intelligible fashion and written in standard English?

Reviewer #1: Yes

Reviewer #3: Yes

6. Review Comments to the Author

Reviewer #1: The authors have addressed my comments. Reviewer 2 raised some very valuable and pertinent comments, but I see the authors have addressed these, backed by referencing (Alu methylation prevents DNA damage and how DNA methylation prevents genomic instability).

Reviewer #3: This paper by Thongsroy and Mutirangura is a revision of a paper that I did not have as a first reader. It attemps to relate the Alu methylation status with hypertension severity using the COBRA approach. It seems that Alu methylation is inversely proportional to diastolic or systolic blood pressure.

I have no major observation concerning the scientific message. The authors took into account the influence of sex of diabetes mellitus. They also analyze the follow-up of patients on a 5 years period (2015-2019), which is one of the most original part of the study. It seems that the age decreases the Alu methylation but not in the pre-hypertensive group, which would warrant an interpretation.

In sum, I would recommend a careful editing of the English by a native English speaker or a professional service. With an adequate discussion about this question of Alu methylation not changed during ageing in the pre-hypertensive group, I’ll recommend publication of this paper.

7. PLOS authors have the option to publish the peer review history of their article (what does this mean?). If published, this will include your full peer review and any attached files.

Reviewer #1: **Yes: **Ronan P. Murphy

Reviewer #3: **Yes: **Daniel Vaiman

---

## [Author Response · Author response to Decision Letter 1]

24 May 2022

Dear Editor,

 We appreciate the reviewers’ comments and have revised our manuscript accordingly. We list our responses in a point-by-point fashion and all the changes in the manuscript text are highlighted in yellow.

Reviewer #1: The authors have addressed my comments. Reviewer 2 raised some very valuable and pertinent comments, but I see the authors have addressed these, backed by referencing (Alu methylation prevents DNA damage and how DNA methylation prevents genomic instability).

Reply: Thank you very much. 

Reviewer #3: This paper by Thongsroy and Mutirangura is a revision of a paper that I did not have as a first reader. It attemps to relate the Alu methylation status with hypertension severity using the COBRA approach. It seems that Alu methylation is inversely proportional to diastolic or systolic blood pressure.

I have no major observation concerning the scientific message. The authors took into account the influence of sex of diabetes mellitus. They also analyze the follow-up of patients on a 5 years period (2015-2019), which is one of the most original part of the study. It seems that the age decreases the Alu methylation but not in the pre-hypertensive group, which would warrant an interpretation.

In sum, I would recommend a careful editing of the English by a native English speaker or a professional service. With an adequate discussion about this question of Alu methylation not changed during ageing in the pre-hypertensive group, I’ll recommend publication of this paper.

Reply: Thank you very much. Here in this version, we have added the sentence “There was no significant difference in prehypertension. Thus, the number of follow-up samples with prehypertension was lower than in the other groups. These limit the use of Alu methylation levels directly as a biomarker for clinical application.” in the discussion part (page 8 lines 230-232). This manuscript was edited by one or more of the the highly qualified native English speaking editors at American Journal Experts (AJE).

---

## [Decision Letter · Decision Letter 2]

2 Jun 2022

The association between Alu hypomethylation and the severity of hypertension

PONE-D-21-37190R2

Dear Dr. Thongsroy,

We’re pleased to inform you that your manuscript has been judged scientifically suitable for publication and will be formally accepted for publication once it meets all outstanding technical requirements.

Kind regards,

H. Hakan Aydin, MD, FAACC

Academic Editor

PLOS ONE

Reviewers' comments:

Reviewer's Responses to Questions

**Comments to the Author**

1. If the authors have adequately addressed your comments raised in a previous round of review and you feel that this manuscript is now acceptable for publication, you may indicate that here to bypass the “Comments to the Author” section, enter your conflict of interest statement in the “Confidential to Editor” section, and submit your "Accept" recommendation.

Reviewer #1: All comments have been addressed

Reviewer #3: All comments have been addressed

2. Is the manuscript technically sound, and do the data support the conclusions?

Reviewer #1: Yes

Reviewer #3: Yes

3. Has the statistical analysis been performed appropriately and rigorously? 

Reviewer #1: Yes

Reviewer #3: Yes

4. Have the authors made all data underlying the findings in their manuscript fully available?

Reviewer #1: Yes

Reviewer #3: Yes

5. Is the manuscript presented in an intelligible fashion and written in standard English?

Reviewer #1: Yes

Reviewer #3: Yes

6. Review Comments to the Author

Reviewer #1: The authors have addressed my comments, and the English editing improves the manuscript. The other reviewers have made valuable comments and edits, which again improve and strengthen the paper.

Reviewer #3: The authors have answered to my concerns. I am satisfied with their improvement of the English. The addition of the limits of their finding in clinics show that they are conscious of the quality and limits of their work.

7. PLOS authors have the option to publish the peer review history of their article (what does this mean?). If published, this will include your full peer review and any attached files.

Reviewer #1: **Yes: **Dr. Ronan P. Murphy

Reviewer #3: No

---

## [Editor Report · Acceptance letter]

30 Jun 2022

PONE-D-21-37190R2 

The association between Alu hypomethylation and the severity of hypertension 

Dear Dr. Thongsroy:

I'm pleased to inform you that your manuscript has been deemed suitable for publication in PLOS ONE. Congratulations! Your manuscript is now with our production department. 

Kind regards, 

on behalf of

Professor H. Hakan Aydin 

Academic Editor

PLOS ONE